# Abrasion Wear Resistance of Polymer Constructional Materials for Rapid Prototyping and Tool-Making Industry

**DOI:** 10.3390/polym12040873

**Published:** 2020-04-10

**Authors:** Janusz Musiał, Serhiy Horiashchenko, Robert Polasik, Jakub Musiał, Tomasz Kałaczyński, Maciej Matuszewski, Mścisław Śrutek

**Affiliations:** 1Faculty of Mechanical Engineering, University of Science and Technology, Kaliskiego 7 Street, 85-789 Bydgoszcz, Poland; janusz.musial@utp.edu.pl (J.M.); jakmus@onet.pl (J.M.); tomasz.kalaczynski@utp.edu.pl (T.K.); maciej.matuszewski@utp.edu.pl (M.M.); 2Khmelnitskiy National University, st Institutskaya st. 11, 29016 Khmelnitskiy, Ukraine; gsl7@ukr.net; 3Faculty of Telecommunications, Computer Science and Electrical Engineering, University of Science and Technology, Kaliskiego 7 Street, 85-789 Bydgoszcz, Poland; mscislaw.srutek@utp.edu.pl

**Keywords:** construction composite, friction resistance, surface state

## Abstract

The original test results of abrasive wear resistance of different type of construction polymer materials were presented and discussed in this article. Tests were made on an adapted test stand (surface grinder for form and finish grinding). Test samples were made of different types of polymer board materials including RenShape®, Cibatool® and phenolic cotton laminated plastic laminate (TCF). An original methodology based on a grinding experimental set-up of abrasion wear resistance of polymer construction materials was presented. Equations describing relations between material type and wear resistance were presented and discussed. Micro and macro structures were investigated and used in wear resistance prediction.

## 1. Introduction

Modern constructional polymers, especially composite materials, are often used in structural applications where the ability to create properties such as stiffness and strength make them attractive compared to traditional engineering materials [1,2,3,4,5]. In addition to structural applications, such materials are also used in applications where both thermal and structural properties are important. Silicon wafers, used in the electronics industry, are such an example. Consequently, coupled thermal–structural analysis of thin structures is becoming increasingly important from a simulation standpoint [6].

Currently, the results of research in the field of materials wear under different conditions of mechanical and thermal loads are dynamically changing [7,8,9]. This is a serious obstacle to the selection or development of constructive technological conditions to increase the durability of products.

The results of many friction and wear mechanisms and phenomena studies of various polymeric materials have been published in recent years. Some of them concerned the determination of various techniques and applications [10,11] for the assessment of wear resistance or the influence of conditions on friction and wear [12,13,14,15]. A significant part of the investigations concerns research on the impact of the material structure, additives or fillers used and coatings on the tribological behavior or wear of cooperating elements [16,17,18,19,20,21,22]. These works relate to the classic approach in determining the wear mechanism, determining wear resistance or friction. In these works, wear is usually determined on the basis of changes in the micro-geometric structure of the surface or based on the object weight loss. This study presents an original method of determining wear resistance in conditions of intensive friction using the geometric structure of the surface, constituted during the test, to determine the tribological properties of materials.

The characteristics of the external heat, lubrication and wear are directly related to the surface geometric structure of cooperating parts. The surface of the part is the outer layer, which differs from the inner part by its structure and other physical properties. The general concept of “surface quality” can be described as set of properties acquired by the surface layer during forming processes. The quality of the surface of the machine parts influences such service properties as the contact fatigue, wear, erosion and crossover resistance. The complex properties of surfaces are formed after processing and determine the concept of “the quality of the surface”. Surface roughness, material structure, physical–chemical–mechanical upper-layer properties and general stress can determine the state (quality) of the surfaces of the machine parts and can be considered in the initial and working (operational) state (Figure 1) [6,23].

The final state of a workpiece surface can be determined by technological processes. Its state (value) is generally satisfactory at the beginning of its cooperation with other surfaces. The nature of the contact of two solids, except for the geometry of the surfaces, it determined by the mechanical, physical and chemical properties of the thin surface layers and their stress states. Such surface layers generally have a different structure and different properties than the materials inside the material. The thickness of upper surface layer ranges from tenths to hundreds of angstroms and, rarely, tenths of a millimeter. The difference between the properties of thin surface layers and the properties of the core is due to the following three main factors [6,24]:The state of the metal atoms in the surface layers, which is different from the state of the atoms in the bulk of the material. The presence of free surface energy and high adsorption activity is a consequence;The sum of mechanical, thermal and physicochemical effects on the metal surface during the final and preliminary processing operations;The sum of repeated cyclic, mechanical, thermal and physicochemical effects on the metal surface in case of friction load in operation.

The outgoing technological terrain quickly disappears in the process of operation. The chemical composition and geometry of the surfaces is completely changed. New surface qualities are formed.

The estimation of geometrical parameters of the surface of machine parts includes the estimation of macro-, micro- and submicron deviations, taking into account the nature and mechanisms of formation of geometric deviations. Deviations distributed into components are caused by machining, internal structure and loading during operation.

Microgeometric deviations can be technological and operational. Technological macro deviations are caused by insufficient precision of the machine, tool, processing modes, temperature stresses and deformations [25,26,27]. Operational macro-deviations usually are caused by uneven wear resulting from misalignment of the moving parts, vibrations and overloads during operation.

The durability of materials can be determined by the following:The combination of material properties and the type (state) of contact surfaces (surface cleanliness, lubrication);The nature of the movement (sliding, rolling, bumps, fluidity);The speed of mutual movement;The load level;The removal of particles that separate or the presence of particles of some other material that complicates friction, etc.

Universal indicator *Ra* used in most cases for measurements, which gives the most complete characteristic with all points of the profile. The value of the average height *Rz* used in case of difficulties is associated with the use of instruments for determining *Ra*. Such characteristics affect the resistance and vibration resistance, as well as the electrical conductivity of materials.

A direct correlation determines the characteristics of the processed surface; the higher the class index, the less important the height of the measured surface is and the better the quality of processing.

The arithmetic mean deviation Ra of the profile, called the arithmetic mean of the absolute values of the profile deviations within the base length *l*—Figure 2 [1,24,28], can be calculated from the following equation:(1)$$Ra=\frac{1}{n}\cdot \overset{n}{\underset{i=1}{\sum }}\left|y_{i}\right|$$

The middle line of the profile is defined so that areas of the projections and troughs of the contour of the profile *F* on both sides of it are equal.

The height of the irregularities of the profile at 10 points *Rz* is the sum of the mean absolute heights of the five largest projections of the profile and the depths of the five largest depressions of the profile within the base length.
(2)$$Rz=\frac{1}{5}\cdot \left(\overset{5}{\underset{i=1}{\sum }}\left|y_{pi}\right|+\overset{5}{\underset{i=1}{\sum }}\left|y_{vi}\right|\right)$$
where $y_{pi}$ is the height of the highest *i*-th protrusion of the profile; and $y_{vi}$ is the depth of the *i*-th largest recess of the profile.

Relative reference length of the profile $t_{p}$ is the reference length of the profile ratio that is equal to the sum of the lengths of the sections. Segments cut off at a given level in the material of the profile by a line equivalent to the average line within the base length to the base length.
(3)$$t_{p}=\overset{n}{\underset{i=1}{\sum }}\frac{b_{i}}{l}$$

The parameter $t_{p}$ characterizes the shape of the profile irregularities and gives the notion of the distribution of the height of the irregularities across the levels of the profile cross section.

The wear and tear processes are uneven and multi-stage. Three periods of wear and tear are observed when parts and assemblies of machines are operating, namely working (1), permanent (normal) wear (2) and catastrophic (emergency) wear (3) (Figure 3). The working out period is characterized by increased wear rate, which is gradually decreasing. The condition of cooperating friction pairs gradually changes, because the initial stages of the process of wear and tear begin to eliminate irregularities on the part’s surface. New relief forms, which are characteristic to specific loading conditions and structural changes of materials, occur. The area of actual contact thus changes, and the coefficient of friction and temperature in the contact zone decrease markedly (Figure 3). As soon as the structure and the relief on the surface of the materials become optimal for these friction conditions, their wear rate decreases to the minimum values.

Period 2 of permanent wear is characterized by the relative constancy of the friction conditions and the wear rate. The coefficient of friction does not change. During this period, a dynamic equilibrium is established in the surface layers of the contacting bodies between the processes of strengthening and weakening, the formation of new structures and their destruction. In the surface layers of materials, the optimum structure formed during the working period and the corresponding relief is preserved during the period of permanent wear, as well as during the beginning of catastrophic wear 3. The wear resistance of machine parts depends essentially on the nature of the relief and the structure formed on the surface of the materials during the working-in period. Therefore, it is important to be able to control the processes of relief formation and structure on the surface of the machine parts in the initial period of wear, that is, during the period of wear.

## 2. Modeling Polymer Constructional Material Wear

An original function based on the analysis of physical processes of resistance of forms to wear and the formation of their resistance was made. This function describes the dynamics of wear for the period of its operation. The dependence of wear W on the amount of abrasion N is described by the following function [28,29,30]:(4)$$W\left(N\right)=\frac{1}{A\cdot b_{0}}\left[\frac{\left(A-N_{0}\right)\cdot N}{N_{0}\cdot \left(A-N_{0}\right)}\right]$$
where $A,\;b_{0},\;N_{0}$ are determined using the least squares method.

The wear rate $\vartheta_{W}$ is a derivative of the function W(N) and has the form
(5)$$\vartheta_{W}\;\left(N\right)=\frac{N_{0}\cdot S}{A\cdot b_{0}\cdot \left(A-N_{0}\right)}\left[\left(A-N_{0}\right)-\left(A-N_{0}\right)\cdot N\right]$$
where A is the number of cycles, and S is the wear area.

Since there is mass loss, $W_{m}=\frac{\Delta m}{S}$.

The wear rate, obtained during the experiment, is determined by the following formula:(6)$$\vartheta_{W}\;\left(N\right)=\frac{\Delta m_{i+1}-\Delta m_{i-1}}{N_{i+1}-N_{i-1}}$$

Since the wear rate depends on the stage of wear, it can be noticed that it decreases with time, but then it can rise again. The wear rate $I_{v}$ is defined as the ratio of wear to the friction path *L*. For composition materials, specific wear by volume can be described by $W_{v}.$
(7)$$W_{v}=\frac{\Delta mk}{\rho \Delta h}$$
where k is a correction coefficient taking into account the wear of the rod, k=0.98; ρ is material density; Δh is the height of the micro-irregularities on the friction surface.

The friction path is determined through the number of friction cycles.
(8)$$L=k\cdot l_{p}\cdot N$$
where $l_{p}$ is the path when executing one cycle.

Permissible wear $I_{v}$ can be determined by setting the number of friction cycles.
(9)$$I_{v}=\frac{W_{v}}{k\cdot l_{p}\cdot N}=\frac{\Delta m\cdot k}{\rho \cdot \Delta h\cdot k\cdot l_{p}\cdot N}$$

The value of the change in the maximum height of surface is ΔRz and the specific volume can be defined as follows:(10)$$W_{v}=\Delta Rz\cdot S\cdot \alpha^{2}$$
where α is the fill factor of the physical area.

The use of polymer constructional materials and composites in the manufacturing industry is rapidly increasing. Compared to traditional metallic engineering materials, polymer constructional materials and composites are lighter and more corrosion resistant, and properties like strength, stiffness and toughness can often be tailored to a specific application. The composite material is typically a laminate of individual layers, where the fibers in each layer are unidirectional.

## 3. Experimental Research Parameters of the Sub-Microrelief

Sub-microgeometry characterizes the type of irregularities, which is a geometric reflection of the structure of surface layers of metal and its imperfections. Submicroscopic relief is considered in the areas of the surface from one to several micrometers. The parameters of the sub-microrelief are quantitatively determined by means of electronic fractography, and in particular the following parameters: the relative surface area covered by films of secondary structures, the area of sections and juvenile areas, the thickness of secondary structures and the height of sub-micro-irregularities. The initial surface is characterized by the presence of a developed sub-relief, which is formed in the process of final technological processing of grinding. The coatings of films of secondary structures have a weakly expressed sub microrelief. When worn in the mode of structural and energy adaptation of the friction surface, the sub-relief of the friction surfaces is covered by secondary structures of its type and is caused mainly by the presence of smoothed areas. The sub-micro-roughness of the patches is negligible [31,32,33,34].

Seven samples of different polymer materials were used for the experiment. Their appearance is presented in Figure 4, and the characteristics are summarized in Table 1. TCF, phenolic cotton laminated plastic (laminate, e.g., tekstolit, tufnol, rezoteks, turbax, nowotex), is commonly used for high durability parts and structures, especially wear resistant ones. RenShape® materials are mostly used for prototyping processes and light structure construction. Cibatool® polymer materials are used for mold making, tool making and modelling.

The number of friction cycles selected for the experiment, calculated from grinding wheel speed and real contact time, was *N* = 10,000. The experiment was carried out for grinding depths of 0.005, 0.075 and 0.01 mm. Grinding wheel characteristic were 1-250x25x76 99A 24 M5B – 50.

To determine the quality of wear resistance, $Q_{R}$, the parameter of relative wear of the material, was determined. It can be calculated from the following equation:(11)$$Q_{R}=\frac{\left(Ra_{i+1}-Ra_{i}\right)}{Ra_{i}}$$
where $Ra_{i}$ is the arithmetic mean deviation in area i and $Ra_{i+1}$ is the arithmetic mean deviation in the next area.

Thus, the wear resistance is determined by the following formula:(12)$$I_{v}=\frac{\Delta m}{\rho \cdot \Delta Ra\cdot l_{p}\cdot N}$$
where Δm is the change of mass in area and ΔRa is surface deviation, $\Delta Ra=\left(Ra_{i+1}-Ra_{i}\right)=Q_{R}Ra_{i}.$

The proposed model allows the wear resistance of composite materials to be predicted when they formed.

For roughness models of the sub-microrelief we can use nonlinear regression, namely
(13)$$Y=a_{0}+b_{1}\cdot Ra+b_{2}\cdot Ra^{2}$$
where the parameter $a_{0}$ is the empirical coefficient that describes the initial surface deviation, and $b_{1},b_{2}$ are empirical coefficients that describe wear sub microrelief.

The results of the experiment are summarized in Table 2.

Results of experimental formation of roughness are shown in Figure 5.

The dependence of Y on the parameter was studied. The parabolic trend was selected at the specification stage. Its parameters were least squares estimated. The statistical significance of the equation was verified using the coefficient of determination and the Fisher test. It was also established that the model parameters were statistically significant.

The following equation is used to describe and assess our proposed model of the roughness based on the correlation of multiple regression:(14)$$Y=a_{0}+a_{1}\cdot Ra+a_{2}\cdot Rz$$
where the parameter $a_{0}$ describes the initial roughness, $a_{1}$ describes wear rate $I_{v}$ and $a_{2}$ describes the change in the maximum height of surface.

Thus, the formation of roughness can describe by the following equation:(15)$$Y=a_{0}+\Delta Rz\frac{N}{S\cdot \alpha^{2}}Ra+\frac{\Delta m\cdot Rz}{\rho \cdot \Delta Ra\cdot l_{p}\cdot N}$$

The results of the experiment are summarized in Table 3.

The partial correlation coefficient measures the correlation of the corresponding features (and *Ra*, *Rz*), provided that the influence of other factors on them is eliminated. Conclusions can be made based on partial coefficients about the validity of including variables in the regression model. If the coefficient is small or insignificant, this means that the relationship between this factor and the effective variable is very weak. Therefore, the factor can be excluded from the model.

## 4. Conclusions

Basic formulas that describe surface roughness were identified. The minimal error of the formula was 2% for sample 1 and the maximum was 8% for sample 7.

Seven samples of different polymer board materials and phenolic cotton laminated plastic laminate were used in experiment. The research conducted allowed us to determine that the best roughness was obtained after sample tests of the sample No. 1, the RenShape BM5273.

The dynamics of surface roughness change during intensive friction was the lowest in the case of test sample 1.

Samples 2 and 7 (TCF and Cibatool® BM5168) also obtained sufficient surface quality after abrasive wear resistance tests. Light structure (porous) materials, namely RenShape® BM5035 (No. 3), Cibatool® BM5005 (No. 4) and RenShape® BM5185 (No. 5), were characterized by reduced resistance to abrasive wear and had traces of internal structure, reflected in the surface roughness. These materials are unsuitable for applications in systems where high friction occurs. High density materials, namely RenShape® BM5273 (No. 1), TCF (No. 2), Cibatool® BM5272 (No. 6) and Cibatool® BM5168 (No. 7), had high abrasive wear resistance, which predisposed these materials to applications in elements subjected to a significant number of friction cycles.

As a result of the calculations, multiple regression equations were obtained for all samples.

The statistical significance of the equations was verified using the coefficient of determination and the Fisher test. They established that in the studied situation, 100% of the total variability of Y was explained by a change in factors Xj. It was also established that the model parameters were statistically significant.

## Figures and Tables

**Figure 1 polymers-12-00873-f001:**
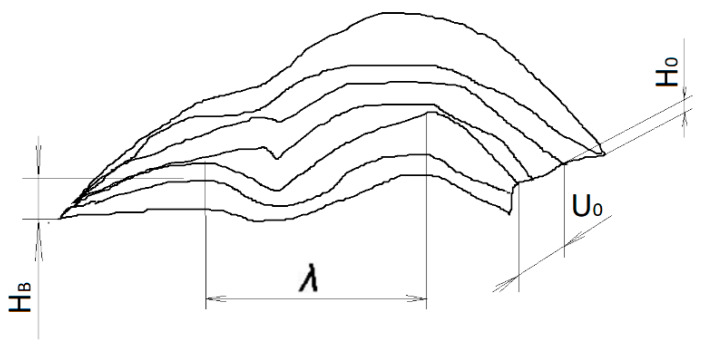
Scheme of surface waviness.

**Figure 2 polymers-12-00873-f002:**
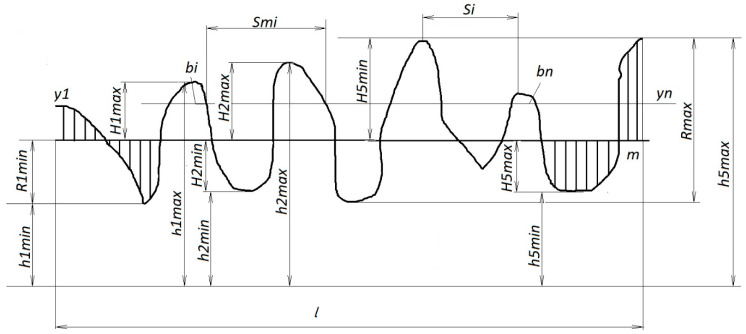
The initial roughness profile.

**Figure 3 polymers-12-00873-f003:**
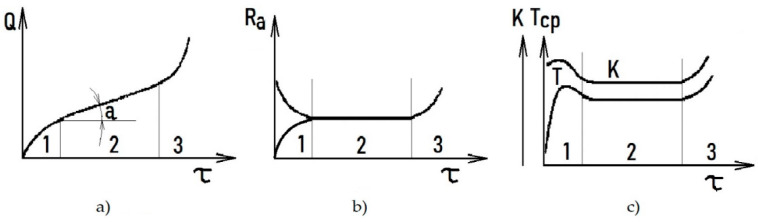
Dependence of the wear rate *Q* (**a**), the height of the micro-irregularities on the friction surface Ra (**b**), the coefficient of friction *K* and the average bulk temperature *T* of the durability elements of the work (**c**); 1,2,3 refer to stages of the wear process.

**Figure 4 polymers-12-00873-f004:**
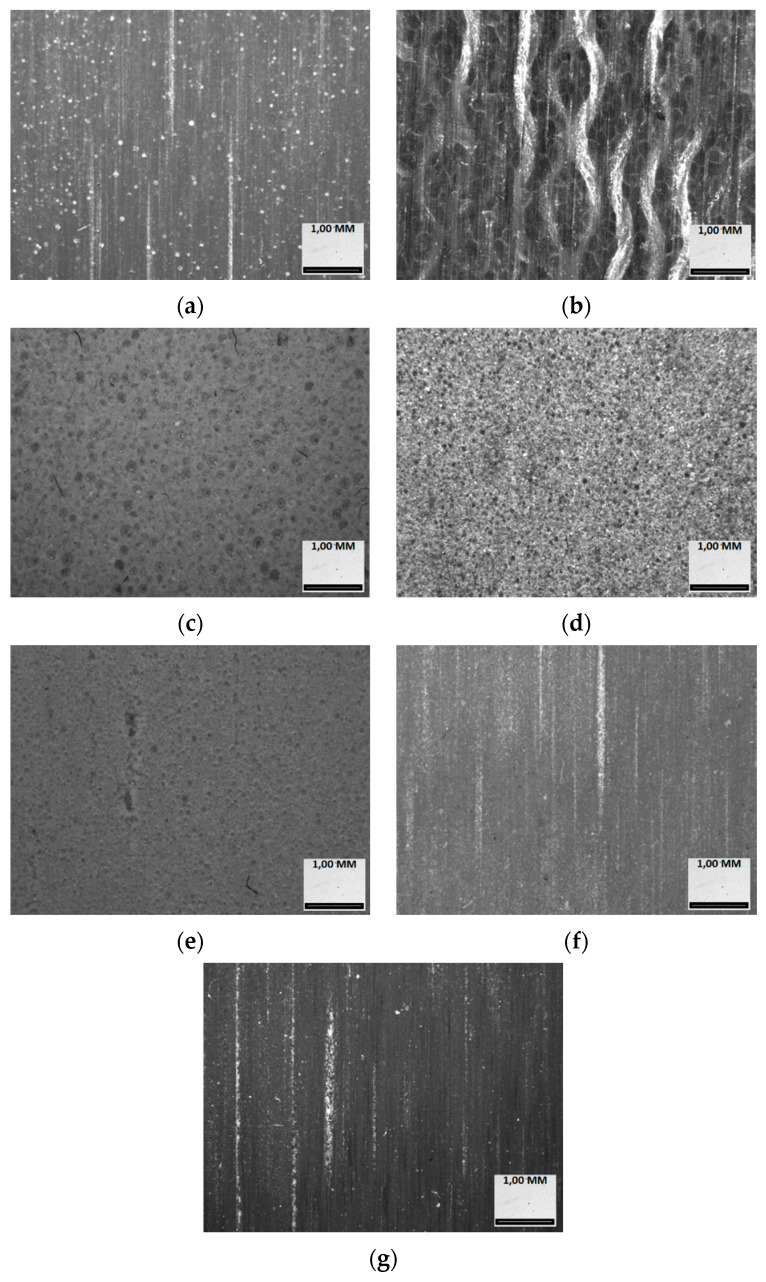
Experimental sample macro structures: (**a**) RenShape® BM5273 (No. 1), (**b**) TCF (Tekstolit, No. 2), (**c**) RenShape® BM5035 (No. 3), (**d**) Cibatool® BM5005 (No. 4), (**e**) RenShape® BM5185 (No. 5), (**f**) Cibatool® BM5272 (No. 6), (**g**) Cibatool® BM5168 (No. 7).

**Figure 5 polymers-12-00873-f005:**
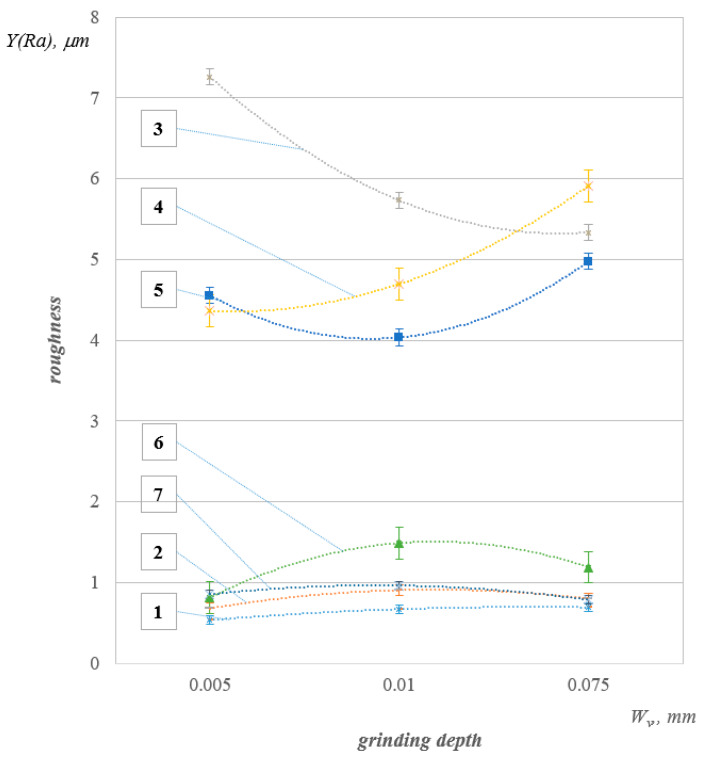
Graphs of experimental formation of roughness.

**Table 1 polymers-12-00873-t001:** Material properties [35,36].

No.	Material	ρ, g/cm^3^	σ, MPa	Shore D
1	RenShape® BM5273	1.4	90	120
2	TCF (Tekstolit)	1.5	80	85
3	RenShape® BM5035	0.45	0,02	48
4	Cibatool® BM5005	0.56	25	68
5	RenShape® BM5185	0.5	15	-
6	Cibatool® BM5272	1.4	80	85
7	Cibatool® BM5168	1.4	90	85

**Table 2 polymers-12-00873-t002:** Results of experiment.

No.	Formula	ΔRa Surface Deviation, µm	Roughness Class
1	$Y=-0.066\cdot Ra^{2}+0.72\cdot Ra+0.12$	0.005264	7
2	$Y=-0.17\cdot Ra^{2}+0.72\cdot Ra+0.12$	0.24025	6
3	$Y=0.566\cdot Ra^{2}-3.22\cdot Ra+9.92$	0.32089	3
4	$Y=0.44\cdot Ra^{2}-0.997\cdot Ra+4.92$	0.227079	4
5	$Y=0.73\cdot Ra^{2}-2.71\cdot Ra+6.53$	0.231453	4
6	$Y=-0.49\cdot Ra^{2}+2.14\cdot Ra-0.84$	0.09569	5
7	$Y=-0.14\cdot Ra^{2}+0.53\cdot Ra+0.46$	0.51174	6

**Table 3 polymers-12-00873-t003:** Results of experiment.

No.	Formula	Roughness Class
1	Y=0.3575+0.4852·Ra−0.00254·Rz	7
2	Y=0.5273−2.1965·Ra+0.1526·Rz	6
3	Y=−0.2199+0.04706·Ra−0.000981·Rz	2
4	Y=0.5758+0.02053·Ra−0.0215·Rz	2
5	Y=0.2429−0.194·Ra+0.02197·Rz	3
6	Y=0.3059+0.2019·Ra−0.05005·Rz	4
7	Y=−26.8319+61.1495·Ra−3.5595·Rz	6

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
