# Peer review of "Abrasion Wear Resistance of Polymer Constructional Materials for Rapid Prototyping and Tool-Making Industry"

_polymers, 2020, doi:10.3390/polym12040873_

Round 1
Reviewer 1 Report
In this paper, the relationship between wear resistance and surface quality of polymer materials is studied, and the prediction formula is put forward. I have the following suggestions.
132-134: Figure 3 should be named by a),b),c)
135,141: For the naming of wear stage, it can be unified without brackets. Replace (3) with 3.
206: There are two pictures of d) in Figure 4, which need to be corrected.
214-216: “The number of friction cycles selected for the experiment, calculated from grinding wheel speed and real contact time, was N=10000. The experiment was carried out for grinding depths; 0.005, 0.075, 0.01mm.” The grinding depth is mentioned in the experimental conditions. But only the number of cycles is mentioned in the condition, and the variables of speed and load are not mentioned. Are these factors consistent in tests on different materials? As we all know, the higher the speed is, the greater the roughness is under the condition of low speed friction cutting.
235-239: Parabola is used to describe the relationship between Y and Wv, but I think it is not accurate to draw a parabola with only three points, at least five points.
255-257: “If the coefficient is small or insignificant, this means that the relationship between this factor and the effective variable is either very weak. Therefore, the factor can be excluded from the model.” I don't fully understand that. If I take No3 as an example, does it mean that Rz can be excluded because 0.000981 is very small? Is this caused by material differences or something else? Can the article analyze this situation?
260: “Basic formulas, that describe surface roughness were identified.” What percentage is the accuracy of the prediction formula? Can the prediction formula compare with a set of real data to give a result?
Author Response
132-134: Figure 3 should be named by a),b),c)
Corrections have been made.
135,141: For the naming of wear stage, it can be unified without brackets. Replace (3) with 3.
Corrections have been made.
206: There are two pictures of d) in Figure 4, which need to be corrected.
Corrections have been made.
214-216: “The number of friction cycles selected for the experiment, calculated from grinding wheel speed and real contact time, was N=10000. The experiment was carried out for grinding depths; 0.005, 0.075, 0.01mm.” The grinding depth is mentioned in the experimental conditions. But only the number of cycles is mentioned in the condition, and the variables of speed and load are not mentioned. Are these factors consistent in tests on different materials? As we all know, the higher the speed is, the greater the roughness is under the condition of low speed friction cutting.
Other experiment conditions were: speed vc 30m/s, feed rate vw 115mm/s, width (traverse) 1,75mm. Conditions were selected in previous experiments to prevent material melting.
235-239: Parabola is used to describe the relationship between Y and Wv, but I think it is not accurate to draw a parabola with only three points, at least five points.
Second-degree curve (quadratic equation) was selected during statistical analysis to explain character of changes. Fig. 5 has been changed.
255-257: “If the coefficient is small or insignificant, this means that the relationship between this factor and the effective variable is either very weak. Therefore, the factor can be excluded from the model.” I don't fully understand that. If I take No3 as an example, does it mean that Rz can be excluded because 0.000981 is very small? Is this caused by material differences or something else? Can the article analyze this situation?
Factors analyses has been given In Conclusions.It was found that in the studied conditions, 100% of the total variability of Y is explained by a change in factors Xj. It was also established that the model parameters are statistically significant.
260: “Basic formulas, that describe surface roughness were identified.” What percentage is the accuracy of the prediction formula? Can the prediction formula compare with a set of real data to give a result?
Added: “the error of the formula is min 2% for sample 1 and max 8% for sample 7”
Reviewer 2 Report
This paper is on the composition effect of polymers for mode and anti-abrasive applications. The wear resistance of different types of polymers with additives was evaluated. The technical content seems falls into the journal. The following suggested modifications are given.
- The article title is not clear. Should the word "model" means "mode" or "mould"? Please check this.
- In the first sentence of Abstract, there is a grammar error. "An original test results of abrasive wear resistance...were..." should be "The original test results of abrasive wear resistance...were...".
- On line 101, "The initial roughness" may be completed as "The initial roughness profile".
- Check Eqn (2) and (3), is "l=1" an error? Should it be "i=1"?
- In Table 1, the unit for stress is wrong. "Mpa" should be corrected as "MPa".
- In Fig. 3, is it possible to have some quantitative information in view of the timeline and wear rate?
- Check the unit in Fig 5. Is it micron or mm?
Author Response
The article title is not clear. Should the word "model" means "mode" or "mould"? Please check this.
The title of the article has been changed to: Abrasion wear resistance of polymer constructional materials for rapid prototyping and tool-making industry
In the first sentence of Abstract, there is a grammar error. "An original test results of abrasive wear resistance...were..." should be "The original test results of abrasive wear resistance...were...".
Corrections have been made.
On line 101, "The initial roughnesst" may be completed as "The initial roughness profile".
Corrections have been made.
Check Eqn (2) and (3), is "l=1" an error? Should it be "i=1"?
Corrections have been made.
In Table 1, the unit for stress is wrong. "Mpa" should be corrected as "MPa".
Corrections have been made.
In Fig. 3, is it possible to have some quantitative information in view of the timeline and wear rate?
Quantitative (specific) information cannot be provided due to the fact that different materials have real (lc) and geometrical (lg) contact length with grinding wheel.
Check the unit in Fig 5. Is it micron or mm?
Corrections have been made.